# Zero-Temperature Equation of State of a Two-Dimensional Bosonic Quantum Fluid with Finite-Range Interaction

**Andrea Tononi** 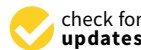

Dipartimento di Fisica e Astronomia "Galileo Galilei" and CNISM, Università di Padova, Via Marzolo 8, I-35131 Padova, Italy; andrea.tononi@phd.unipd.it

**Abstract:** We derive the two-dimensional equation of state for a bosonic system of ultracold atoms interacting with a finite-range effective interaction. Within a functional integration approach, we employ a hydrodynamic parameterization of the bosonic field to calculate the superfluid equations of motion and the zero-temperature pressure. The ultraviolet divergences, naturally arising from the finite-range interaction, are regularized with an improved dimensional regularization technique.

**Keywords:** Bose–Einstein condensation; ultracold atoms; finite-range; equation of state; two-dimensional

## 1. Introduction

A fluid perspective on the study of bosonic gases made of ultracold atoms may have originated in the pioneering work of Landau [1], who correctly described the superfluid behavior of He-4 as observed in his experiments [2]. From those years, several technical advances allowed a precise experimental control of the atomic gases, culminating in the achievement of Bose–Einstein condensation in 1995 [3–5]. Despite the rich phenomenology deriving from the variety of different trapping potentials [6], the theoretical and experimental study of uniform gases gives a fundamental insight into the intrinsic properties of the condensates. Particularly interesting is the case of two spatial dimensions, in which quantum and thermal fluctuations play a fundamental role [7,8], justifying the necessity of a beyond mean field theory. Historical results for a two-dimensional Bose gas were obtained by Schick, who calculated the thermodynamics of a gas of hard-spheres [9], improved by Popov derivation of the equation of state for a weakly-interacting superfluid [10]. Recent works provide an extension of the Popov approach [11], while the nonuniversal corrections to the equation of state in $D = 2$, arising from a finite-range interaction between the atoms, have been studied [12,13]. Thank to the tunability of interparticle interactions, it is possible to investigate the static and dynamical properties of homogeneous quantum fluids in $D$-spatial dimensions in regimes where finite-range corrections are relevant [14–16]. In this work, we provide an alternative derivation of the zero-temperature equation of state by adopting an explicit superfluid parametrization of the bosonic field. In particular, we develop an improved dimensional regularization technique to regularize the zero-temperature pressure of a bosonic quantum fluid, whose particles interact with a finite-range interaction.

## 2. Zero-Temperature Equation of State of a Two-Dimensional Bosonic Quantum Fluid

### 2.1. Superfluid Parametrization of the Bosonic Field

We introduce the Euclidean Lagrangian $\mathcal{L}$ of a uniform quantum fluid of bosonic particles with mass $m$, described by the complex field $\psi(\vec{r}, t)$, namely [17]

$$\mathcal{L} = \bar{\psi}(\vec{r}, \tau) \left( \hbar \partial_\tau - \frac{\hbar^2 \nabla^2}{2m} - \mu \right) \psi(\vec{r}, \tau) + \frac{1}{2} \int d^D r' \, |\psi(\vec{r}', \tau)|^2 \, V(|\vec{r} - \vec{r}'|) \, |\psi(\vec{r}, \tau)|^2, \qquad (1)$$

where $\hbar$ is Planck's constant, $\mu$ is the chemical potential, and we suppose that the particles interact with the isotropic two-body potential $V(|\vec{r} - \vec{r}'|)$. The imaginary time $\tau$ is introduced for uniformity with a functional integration approach, but real time $t$ can be recovered at any moment by performing the Wick rotation $\tau \to it$. According to the least action principle, the Euler–Lagrange equations of the system are obtained as the functional derivative of the action $S[\bar{\psi}, \psi]$, which reads

$$S[\bar{\psi}, \psi] = \int_0^{\beta\hbar} d\tau \int_{L^D} d^D r \, \mathcal{L} \qquad (2)$$

where $L^D$ is the volume in $D$ dimensions containing the particles and $\beta = 1/(k_B T)$, with $T$ the absolute temperature, and $k_B$ the Boltzmann constant. Until the end of this paper, for reasons connected to the dimensional regularization of the final results, we will not explicitly fix the spatial dimension to $D = 2$. The minimization of the action of Equation (2) gives the Gross–Pitaevski equation for the complex field $\psi(\vec{r}, \tau)$, which constitutes the macroscopic wavefunction of the condensate [18]. In this work, however, we adopt a superfluid perspective through the following phase-amplitude parametrization of the bosonic field [19]

$$\psi(\vec{r}, \tau) = \sqrt{\rho(\vec{r}, \tau)} \, e^{i\theta(\vec{r}, \tau)}, \qquad (3)$$

where $\rho(\vec{r}, \tau) = |\psi(\vec{r}, \tau)|^2$ is a real field describing the system local density and $\theta(\vec{r}, \tau)$ is the phase field, which must be included because of the complex nature of the order parameter $\psi(\vec{r}, \tau)$. This field transformation allows us to introduce the superfluid velocity $\vec{v}_s$, which is proportional to the gradient of the phase, namely

$$\vec{v}_s = \frac{\hbar}{m} \vec{\nabla} \theta. \qquad (4)$$

We emphasize that the phase field $\theta(\vec{r}, \tau)$ is defined in the compact interval $[0, 2\pi]$ and is therefore periodic of $2\pi$: this fact constitutes the origin of many topological phenomena in condensed-matter physics. Indeed, here we focus on two-dimensional systems, where the singularities of the phase field—the vortices—are responsible for the Berezinskii–Kosterlitz–Thouless (BKT) transition [20,21]. However, in the following, we will study only the zero-temperature properties, for which the vortex–antivortex phenomenology does not play a fundamental role and can be neglected. We will then assume that the domain of definition of the phase field $\theta(\vec{r}, \tau)$ can be extended to $\mathbb{R}$ and that its spatial and time derivatives are well defined everywhere. In this case, the superfluid flow is irrotational, thus it has zero vorticity

$$\vec{\nabla} \times \vec{v}_s = 0. \qquad (5)$$

We now substitute the parametrization of Equation (3) in the Lagrangian (1), obtaining

$$\mathcal{L} = -\mu \rho + i\hbar \rho \partial_\tau \theta + \frac{\hbar^2}{8m\rho} (\nabla \rho)^2 + \frac{\hbar^2 \rho}{2m} (\nabla \theta)^2 + \frac{1}{2} \int d^D r' \, \rho(\vec{r}', \tau) \, V(|\vec{r} - \vec{r}'|) \, \rho(\vec{r}, \tau), \qquad (6)$$

where we omit for simplicity the dependence of the fields on their coordinates $(\vec{r}, \tau)$. The minimization of the action (2), which now becomes a functional of $\rho$ and $\theta$: $S = S[\rho, \theta]$, leads to the Euler–Lagrange equations for these fields. Recovering the real time $t \to -i\tau$, we get the hydrodynamic equations

$$\frac{\partial \rho}{\partial t} + \vec{\nabla} \cdot (\rho \vec{v}_s) = 0 \tag{7}$$

and

$$m \frac{\partial \vec{v}_s}{\partial t} + \vec{\nabla} \left( \frac{1}{2} m v_s^2 - \mu + \int d^D r' \, \rho(\vec{r}', \tau) \, V(|\vec{r} - \vec{r}'|) - \frac{\hbar^2}{2m\rho^{1/2}} \nabla^2(\rho^{1/2}) \right) = 0, \tag{8}$$

which are the continuity equation and the equation of motion of a superfluid with velocity $\vec{v}_s$ and number density $\rho$. Notice that, inside the parenthesis, Equation (8) contains the quantum pressure term: if the density $\rho$ of the fluid is slowly varying and this contribution can be neglected, Equation (8) reduces to the familiar Euler equation of motion for an irrotational fluid without viscosity. For a self-consistent derivation of these equations from the Gross–Pitaevski equation we refer the reader to Reference [22].

*2.2. Zero-Temperature Equation of State*

We now adopt the superfluid parametrization of Equation (3) to derive the zero-temperature equation of state of the quantum fluid, namely a relation at $T = 0$ between the pressure $P$ and the chemical potential $\mu$. For the finite-range interaction, an explicit implementation of this relation will be given in the next section.

In the grand canonical ensemble, we calculate the pressure of the bosonic fluid as $P = -\Omega/L^D$, where $\Omega$ is the grand potential

$$\Omega = -\frac{1}{\beta} \ln(\mathcal{Z}) \tag{9}$$

and $\mathcal{Z}$ is the grand canonical partition function, which, within a functional integration perspective, can be calculated as

$$\mathcal{Z} = \int \mathcal{D}[\rho, \theta] \, e^{-\frac{S[\rho, \theta]}{\hbar}}. \tag{10}$$

To perform the explicit functional integration of the Lagrangian (6), we rewrite the local density field $\rho(\vec{r}, \tau)$ as

$$\rho(\vec{r}, \tau) = \rho_0 + \delta\rho(\vec{r}, \tau), \tag{11}$$

where $\rho_0$ is the condensate density of the system in the broken-symmetry phase and $\delta\rho(\vec{r}, \tau)$ is a real field describing the density fluctuations.

We substitute the field transformation (11) in the Lagrangian of Equation (6), obtaining

$$\mathcal{L} = -\mu\rho_0 - \mu \, \delta\rho + i\hbar\rho_0 \partial_\tau \theta + \frac{\hbar^2}{8m\rho_0} (\nabla\delta\rho)^2 + \frac{\hbar^2\rho_0}{2m} (\nabla\theta)^2 +$$
$$\frac{1}{2} \int d^D r' \, V(|\vec{r} - \vec{r}'|)(\rho_0^2 + \delta\rho(\vec{r}, \tau) + \delta\rho(\vec{r}', \tau) + \rho(\vec{r}, \tau)\delta\rho(\vec{r}', \tau)), \tag{12}$$

where we keep only terms up to the second order in the fluctuation fields $\delta\rho(\vec{r}, \tau)$ and $\theta(\vec{r}, \tau)$, thus making a Gaussian (one-loop) approximation.

Considering the Lagrangian of Equation (12) inside the action $S$, which now becomes the functional $S = S[\delta\rho, \theta]$, it is particularly convenient to express it in terms of the Fourier series of the fluctuation fields, namely

$$
\begin{aligned}
\delta\rho(\vec{r}, \tau) &= \frac{1}{\sqrt{L^D}} \sum_{\vec{k}\,\omega_n} e^{i\vec{k}\cdot\vec{r}} e^{-i\omega_n\tau} \, \delta\rho(\vec{k}, \omega_n) \\
\theta(\vec{r}, \tau) &= \frac{1}{\sqrt{L^D}} \sum_{\vec{k}\,\omega_n} e^{i\vec{k}\cdot\vec{r}} e^{-i\omega_n\tau} \, \theta(\vec{k}, \omega_n) \\
\delta\rho(\vec{k}, \omega_n) &= \frac{1}{\beta\hbar\sqrt{L^D}} \int_0^{\beta\hbar} d\tau \int_{L^D} d^D r \, e^{-i\vec{k}\cdot\vec{r}} e^{i\omega_n\tau} \, \delta\rho(\vec{r}, \tau) \\
\theta(\vec{k}, \omega_n) &= \frac{1}{\beta\hbar\sqrt{L^D}} \int_0^{\beta\hbar} d\tau \int_{L^D} d^D r \, e^{-i\vec{k}\cdot\vec{r}} e^{i\omega_n\tau} \, \theta(\vec{r}, \tau),
\end{aligned}
\tag{13}
$$

where $\omega_n = 2\pi n/(\beta\hbar)$ are the bosonic Matsubara frequencies. Notice that, since we are supposing that the phase field $\theta(\vec{r}, \tau)$ is defined on $\mathbb{R}$, its Fourier components are non-numerable and can assume continuous values, thus they can be treated like ordinary functional integral variables. The action in the Fourier space is obtained by simply substituting these expressions in $S$ and using the definition of the $D + 1$-dimensional delta function. Moreover, we also substitute the Fourier series $\tilde{V}(k)$ of the real space interaction potential and we define with $g_0$ the zero-range interaction strength $g_0 = \tilde{V}(k = 0)$. In this way, the action can be rewritten as the sum of two contributions

$$
S = S_0 + S_g.
\tag{14}
$$

The first is the action of the homogeneous system $S_0$, namely

$$
S_0 = \beta\hbar L^D\left( -\mu\rho_0 + \frac{1}{2}g_0\rho_0^2 \right),
\tag{15}
$$

which does not depend on the functional integration variables: using Equations (9) and (10), one can employ $S_0$ to calculate $\Omega_0$, the mean field contribution to the grand potential

$$
\Omega_0 = \left( -\mu\rho_0 + \frac{1}{2}g_0\rho_0^2 \right)L^D.
\tag{16}
$$

The second contribution to the action $S$ is the Gaussian action $S_g$, which is given by

$$
S_g = \beta\hbar \sum_{\vec{k}\,\omega_n} \left[ \frac{\hbar^2 k^2 \rho_0}{2m}\theta(k)\theta(-k) + \left( \frac{\hbar^2 k^2}{8m\rho_0} + \frac{\tilde{V}(k)}{2} \right)\delta\rho(k)\delta\rho(-k) + \hbar\omega_n\theta(k)\delta\rho(-k) \right],
\tag{17}
$$

where, for simplicity of notation, we define $\delta\rho(\pm k) = \delta\rho(\pm\vec{k}, \pm\omega_n)$ and $\theta(\pm k) = \theta(\pm\vec{k}, \pm\omega_n)$. Since $S_g$ is quadratic in the fluctuation fields $\delta\rho(k)$ and $\theta(k)$, one can rewrite it in the following matricial form:

$$
S_g = \frac{\hbar}{2} \sum_{\vec{k}\,\omega_n} \begin{pmatrix} \theta(k) & \theta(-k) & \delta\rho(k) & \delta\rho(-k) \end{pmatrix} \mathbf{M}(k) \begin{pmatrix} \theta(k) \\ \theta(-k) \\ \delta\rho(k) \\ \delta\rho(-k) \end{pmatrix},
\tag{18}
$$

where $\mathbf{M}(k)$, the inverse of the propagator, is the $4 \times 4$ matrix

$$\mathbf{M}(k) = \beta \begin{pmatrix} 0 & \frac{\hbar^2 k^2 \rho_0}{m} & 0 & \hbar \omega_n \\ \frac{\hbar^2 k^2 \rho_0}{m} & 0 & -\hbar \omega_n & 0 \\ 0 & -\hbar \omega_n & 0 & \frac{\hbar^2 k^2}{4m\rho_0} + \tilde{V}(k) \\ \hbar \omega_n & 0 & \frac{\hbar^2 k^2}{4m\rho_0} + \tilde{V}(k) & 0 \end{pmatrix}. \tag{19}$$

The functional integral of the real fluctuation fields $\theta(k)$ and $\delta\rho(k)$ can be performed explicitly [23], obtaining the corresponding Gaussian grand canonical partition function $\mathcal{Z}_g$ as

$$\mathcal{Z}_g = \prod_{\substack{\vec{k}\, \omega_n \\ k_z > 0}} [\det \mathbf{M}(k)]^{-1/2}, \tag{20}$$

which, considering the definition of the grand potential of Equation (9), leads to the Gaussian contribution to the grand potential

$$\Omega_g = \frac{1}{2\beta} \sum_{\vec{k}\, \omega_n} \ln[\beta^2(\hbar^2 \omega_n^2 + E_k^2)]. \tag{21}$$

Here, we find the gapless excitation spectrum $E_k$ of the quantum fluid in the form

$$E_k = \sqrt{\frac{\hbar^2 k^2}{2m}\left(\frac{\hbar^2 k^2}{2m} + 2\rho_0 \tilde{V}(k)\right)}, \tag{22}$$

where, within a perturbative approach, $\rho_0$ is determined by the saddle point condition $\partial\Omega_0/\partial\rho_0 = 0$, which leads to

$$\rho_0 = \frac{\mu}{g_0} \tag{23}$$

and whose substitution in the excitation spectrum gives $E_k^B$, the renowned Bogoliubov spectrum [24]

$$E_k^B = \sqrt{\frac{\hbar^2 k^2}{2m}\left(\frac{\hbar^2 k^2}{2m} + 2\mu \frac{\tilde{V}(k)}{g_0}\right)}. \tag{24}$$

The sum over the Matsubara frequencies $\omega_n$ in the Gaussian grand potential of Equation (21) can be performed according to the prescriptions described in the Appendix A, obtaining the grand potential as the sum of three contributions

$$\Omega = \Omega_0 + \Omega_g^{(0)} + \Omega_g^{(T)}, \tag{25}$$

where $\Omega_0 = -L^D \mu^2/(2g_0)$ due to Equations (16) and (23), and

$$\Omega_g^{(0)} = \frac{1}{2} \sum_{\vec{k}} E_k^B \tag{26}$$

is the zero-temperature Gaussian grand potential encoding quantum fluctuations, while

$$\Omega_g^{(T)} = \frac{1}{\beta} \sum_{\vec{k}} \ln(1 - e^{-\beta E_{\vec{k}}^B}) \tag{27}$$

is the finite-temperature Gaussian grand potential, encoding thermal fluctuations. Finally, we explicitly write the zero-temperature equation of state, namely, we calculate the pressure as the opposite of the grand potential of Equation (25) at $T = 0$:

$$P(\mu, T = 0) = \frac{\mu^2}{2g_0} - \frac{1}{2L^D} \sum_{\vec{k}} E_k^B. \tag{28}$$

In the thermodynamic limit of $L \to \infty$, the sum over $\vec{k}$ can be rewritten as a $D$-dimensional integral in momentum space $(2\pi/L)^D \sum_{\vec{k}} = \int d^D k$, and, substituting again the Bogoliubov spectrum (24),the equation of state becomes

$$P(\mu, T = 0) = \frac{\mu^2}{2g_0} - \frac{1}{2} \int \frac{d^D k}{(2\pi)^D} \sqrt{\frac{\hbar^2 k^2}{2m} \left( \frac{\hbar^2 k^2}{2m} + 2\mu \frac{\tilde{V}(k)}{g_0} \right)}, \tag{29}$$

where the integral can be calculated after the explicit choice of $\tilde{V}(k)$.

### 2.3. Explicit Implementation for Finite-Range Interaction

We now provide an explicit implementation of the zero-temperature equation of state (29) for a bosonic quantum fluid of particles interacting with the finite-range effective interaction

$$\tilde{V}(k) = g_0 + g_2 k^2, \tag{30}$$

where $g_0 = \tilde{V}(k = 0)$ is the usual zero-range interaction coupling, and

$$g_2 = \frac{1}{2} \int d^2 r \, r^2 \, V(|\vec{r}|) \tag{31}$$

is the first nonzero correction in the gradient expansion of an isotropic interaction potential $V(|\vec{r}|)$. At zero temperature, we expect the finite-range corrections to the equation of state to be detectable, but small with respect to the zero-range result of Reference [25]. By using scattering theory in two spatial dimensions, these couplings can be linked with the s-wave scattering length $a_s$ and the characteristic range $R$ of the real interatomic two-body interaction [12,26,27]

$$g_0 = \frac{4\pi\hbar^2}{m|\ln(na_s^2)|}, \qquad g_2 = \frac{\pi\hbar^2 R^2}{m|\ln(na_s^2)|}, \tag{32}$$

where $n$ is the number density of the system in $D = 2$.

The equation of state (29) becomes, with the finite-range interaction of Equation (30)

$$P(\mu, T = 0) = \frac{\mu^2}{2g_0} + P_g^{(0)}, \tag{33}$$

where we define the zero temperature Gaussian pressure $P_g^{(0)}$ as

$$P_g^{(0)} = -\frac{1}{2} \int \frac{d^D k}{(2\pi)^D} \sqrt{\frac{\hbar^2 k^2}{2m} \left( \frac{\hbar^2 k^2}{2m} \lambda + 2\mu \right)}, \tag{34}$$

with

$$\lambda = 1 + \frac{4m}{\hbar^2} \frac{\mu}{g_0} g_2. \tag{35}$$

Since the integrand function depends only on the modulus of the momentum $|\vec{k}|$, we rewrite the integral in $P_g^{(0)}$ using $D$-dimensional spherical coordinates, namely

$$P_g^{(0)} = -\frac{S_D}{2(2\pi)^D} \int_0^{+\infty} dk \, k^{D-1} \sqrt{\frac{\hbar^2 k^2}{2m} \left( \frac{\hbar^2 k^2}{2m} \lambda + 2\mu \right)}, \tag{36}$$

where $S_D = 2\pi^{D/2}/\Gamma[D/2]$ is the solid angle in $D$-dimensions and $\Gamma[D/2]$ is the Euler Gamma function. In order to integrate this equation, we introduce the adimensional variable $t = \hbar^2 k^2 \lambda/(4m\mu)$, obtaining

$$P_g^{(0)} = -\frac{\mu}{\lambda^{1/2}\Gamma[D/2]} \left( \frac{m\mu}{\pi\hbar^2\lambda} \right)^{D/2} \int_0^{+\infty} dt \, t^{\frac{D-1}{2}} (1+t)^{1/2}. \tag{37}$$

As a consequence of the substitution of the real interatomic potential with an effective interaction, the zero-temperature Gaussian pressure $P_g^{(0)}$ is ultraviolet divergent. In our framework, an efficient way to regularize $P_g^{(0)}$ is constituted by the technique of dimensional regularization [28]. The basic idea of this approach is to rewrite a diverging integral in terms of the Euler beta and gamma functions, whose integral representation for $x, y, z > 0$ is given by

$$B(x,y) = \int_0^{+\infty} dt \, \frac{t^{x-1}}{(1+t)^{x+y}}, \tag{38}$$

$$\Gamma(z) = \int_0^{+\infty} dt \, t^{z-1} \, e^{-z}. \tag{39}$$

Thanks to the properties $B(x,y) = \Gamma(x)\Gamma(y)/\Gamma(x+y)$ and $\Gamma[z+1] = z\,\Gamma[z]$, one can extend the domain of definition of the gamma and beta functions by analytic continuation of their arguments $x, y, z$ also to negative values, which usually appear in many physical problems. However, despite this, the dimensional regularization procedure can be successfully used to regularize many ultraviolet diverging integrals, in our peculiar two-dimensional case the procedure described above would lead to a result containing the gamma function evaluated for negative integer values, which is again a diverging quantity. To avoid this residual divergence, we extend the dimensions of the system to the complex value $\mathcal{D} = D - \varepsilon$, and we formally perform the integration of Equation (37). We obtain

$$P_g^{(0)} = \frac{\kappa^\varepsilon}{2} \left( \frac{\mu}{\pi\lambda} \right)^{(\mathcal{D}+1)/2} \left( \frac{m}{\hbar^2} \right)^{\mathcal{D}/2} \frac{\Gamma[(D-\varepsilon+1)/2]\,\Gamma[(\varepsilon-D-2)/2]}{\Gamma[(D-\varepsilon)/2]}, \tag{40}$$

in which the wavevector $\kappa$ is introduced for dimensional reasons. Notice how in $D = 2$ and for $\varepsilon = 0$ the Gaussian pressure is still divergent. To regularize it, we rely on the following small-$\varepsilon$ expansion of the gamma function [29]

$$\Gamma(-n+\varepsilon) = \frac{(-1)^n}{n!} \left[ \frac{1}{\varepsilon} + \Psi(n+1) + \frac{\varepsilon}{2} \left( \frac{\pi^2}{3} + \Psi(n+1)^2 - \Psi'(n+1) \right) + o(\varepsilon^2) \right], \tag{41}$$

where $\Psi(n+1)$ is the digamma function and $\Psi'(n+1)$ is its derivative. Moreover, we express the exponentiation of a generic coefficient $x^\varepsilon$ for $\varepsilon \to 0$ as

$$x^\varepsilon = \exp(\varepsilon \ln(x)) \sim_{\varepsilon\to 0} 1 + \varepsilon \ln(x) + o(\varepsilon^2). \tag{42}$$

With this recipe, the Gaussian pressure $P_g^{(0)}$ in $D = 2$ gives

$$P_g^{(0)} = \frac{m\mu^2}{2\pi^{3/2}\hbar^2\lambda^{3/2}} \left[ \frac{\pi^{1/2}}{2}\frac{1}{\varepsilon} + \frac{\pi^{1/2}}{8}(\ln(16) - 2\gamma - 1) + \frac{\pi^{1/2}}{4}\ln\left(\frac{\pi\lambda\hbar^2\kappa^2}{m\mu}\right) + o(\varepsilon) \right], \quad (43)$$

where $\gamma \approx 0.55722$ is the Euler–Mascheroni constant. Finally, we delete the $o(\varepsilon^{-1})$ divergence in the square bracket [30] and we rewrite the zero-temperature equation of state $P(\mu, T = 0)$ of Equation (29) as

$$P(\mu, T = 0) = \frac{m\mu^2}{8\pi\hbar^2\lambda^{3/2}} \left[ \ln\left(\frac{\epsilon_0}{\mu}\lambda\right) - \frac{1}{2} \right], \quad (44)$$

where we define the energy cutoff $\epsilon_0$ as

$$\epsilon_0 = \frac{4\pi\hbar^2\kappa^2}{m \exp(\gamma - \frac{4\pi\hbar^2\lambda^{3/2}}{mg_0})}. \quad (45)$$

The equation of state (44) improves the one derived for bosons with a zero-range interaction [10] by Popov, whose result can be reproduced by setting $\lambda = 1$, i.e., $g_2 = 0$. We emphasize that, with a precise tuning of the interparticle interaction (see Reference [12] for a detailed discussion), the finite-range corrections derived within our Gaussian approximation become larger than the zero-range beyond-Gaussian ones obtained by Mora and Castin [31]. For weakly-interacting bosons with $na_s^2 \ll 1$, where $a_s$ is the two-dimensional s-wave scattering length, we expect that the nonuniversal corrections of Equation (44) arise for $R \geq a_s$, where $R$ is the characteristic range of the interaction. In this intermediate regime, the neglection of higher order terms in the gradient expansion of Equation (30) is justified but, at the same time, the finite-range contributions are of comparable size to the zero-range ones.

## 3. Conclusions

In this work, we derive the two-dimensional zero-temperature equation of state for a bosonic quantum fluid with a generic isotropic interaction. The superfluid perspective is emphasized by performing the Gaussian functional integration within a phase-amplitude parametrization of the complex order parameter. For a system with zero-range interaction, we reproduce the classical result by Popov. Nonetheless, we apply a novel dimensional regularization recipe to reproduce the nonuniversal corrections for a finite-range interaction potential. We expect, with a fine-tuning of the experimental interaction parameters, the finite-range correction to produce sizable corrections to the thermodynamics of the weakly-interacting superfluid. Our derivation of the zero-temperature equation of state is also valid for other interparticle interactions. In particular, the previous results can be extended for a quasi-two-dimensional system of dipolar bosons whose polarization direction is perpendicular to the plane of confinement. For a generic orientation, however, it is necessary to consider the dependence of the interaction on the in-plane angle between the particles and to include it consistently in the dimensional regularization procedure.

**Funding:** This research received no external funding.

**Acknowledgments:** The author thanks Luca Salasnich and Alberto Cappellaro for useful discussions and suggestions.

**Conflicts of Interest:** The author declares no conflict of interest.

## Appendix A

We illustrate here the procedure to calculate the summation over the bosonic Matsubara frequencies $\omega_n$, which are defined as

$$\omega_n = \frac{2\pi n}{\beta\hbar}, \tag{A1}$$

where $n \in \mathbb{Z}$ are integer numbers. The most common sum that one has to perform is in the form

$$I[\xi_{\vec{k}}] = \frac{1}{2\beta} \sum_{n=-\infty}^{+\infty} \ln[\beta^2(\hbar^2\omega_n^2 + \xi_{\vec{k}}^2)]. \tag{A2}$$

Using the properties of the logarithm and considering that the summation involves all $n \in \mathbb{Z}$ integers, both positive and negative, $I[\xi_{\vec{k}}]$ can also be rewritten in the useful form

$$I[\xi_{\vec{k}}] = \frac{1}{\beta} \sum_{n=-\infty}^{+\infty} \ln[\beta(-i\hbar\omega_n + \xi_{\vec{k}})]. \tag{A3}$$

Taking the derivative of $I[\xi_{\vec{k}}]$ with respect to $\xi_{\vec{k}}$ in Equation (A2), we get

$$\frac{\partial I[\xi_{\vec{k}}]}{\partial \xi_{\vec{k}}} = \frac{1}{\beta} \sum_{n=-\infty}^{+\infty} \frac{\xi_{\vec{k}}}{\hbar^2\omega_n^2 + \xi_{\vec{k}}^2}. \tag{A4}$$

In the zero temperature limit, the difference

$$\Delta\omega = \omega_n - \omega_{n-1} = \frac{2\pi}{\beta\hbar} \xrightarrow[\beta \gg 1]{} d\omega \tag{A5}$$

becomes infinitesimal and we can substitute the sum over $n$ with an integral over $\omega$, obtaining

$$\frac{\partial I[\xi_{\vec{k}}]}{\partial \xi_{\vec{k}}} = \frac{1}{\beta} \int_{-\infty}^{+\infty} d\omega \, \frac{\beta\hbar}{2\pi} \frac{\xi_{\vec{k}}}{\hbar^2\omega^2 + \xi_{\vec{k}}^2} = \frac{1}{2}, \tag{A6}$$

which is the zero-temperature contribution to $I[\xi_{\vec{k}}]$. If the temperature is relatively low, but non-zero, we cannot substitute the sum in Equation (A4) with an integral, but we can rewrite it as

$$\frac{\partial I[\xi_{\vec{k}}]}{\partial \xi_{\vec{k}}} = \frac{\beta\xi_{\vec{k}}}{(2\pi)^2} \sum_{n=-\infty}^{+\infty} \frac{1}{n^2 + \left(\frac{\beta\xi_{\vec{k}}}{2\pi}\right)^2} \tag{A7}$$

and, using the identity

$$\sum_{n=0}^{+\infty} \frac{1}{n^2 + a^2} = \frac{1 + \pi a \, \coth(\pi a)}{2a^2}, \tag{A8}$$

we obtain

$$\frac{\partial I[\xi_{\vec{k}}]}{\partial \xi_{\vec{k}}} = \frac{1}{2} \coth\left(\frac{\beta\xi_{\vec{k}}}{2}\right) = \frac{1}{2} + \frac{1}{e^{\beta\xi_{\vec{k}}} - 1}. \tag{A9}$$

We integrate this equation on $\xi_{\vec{k}}$, and setting the arbitrary constant resulting from the indefinite integral to zero (it is not dependent on physical parameters), we finally obtain the result of the summation over the Matsubara frequencies

$$I[\xi_{\vec{k}}] = \frac{\xi_{\vec{k}}}{2} + \frac{1}{\beta} \ln(1 - e^{-\beta\xi_{\vec{k}}}), \tag{A10}$$

which is used in this article to obtain Equation (21).

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
