# Peer review of "Zero-Temperature Equation of State of a Two-Dimensional Bosonic Quantum Fluid with Finite-Range Interaction"

_condensedmatter, doi:10.3390/condmat4010020_

Round 1

Reviewer 1 Report

The author derives equation of state for a 2D bosonic quantum fluid with finite-range interactions, assuming no vortices are present. The system is assumed to be homogeneous (no trap) and at T=0. Using a Madelung decomposition, the author rewrites system's Lagrangian in a suitable form and derives the corresponding Euler-Lagrange equations. Furthermore, using a path integral approach in a grand-canonical ensemble setup, the authors obtains equation of state in terms of the scattering length and the finite interaction range.

The paper is well written and its results are interesting, although the language should be slightly polished. It would be nice to see some discussion on the possibility to use this approach for bosonic systems with long-range interaction, such as the dipole-dipole interaction.

Author Response

The author thanks the referee for kindly reviewing the paper and for considering the manuscript well written and interesting.

The referee writes: the language should be slightly polished.

Reply: After a careful revision of the manuscript, I tried to improve the language and to remove unusual linguistic forms. The main corrections are highlighted in red.

The referee writes: It would be nice to see some discussion on the possibility to use this approach for bosonic systems with long-range interaction, such as the dipole-dipole interaction.

Reply: A discussion about extending this work to the long-range dipole-dipole interatomic interaction has been added in the conclusions. Essentialy, the equation of state can be easily calculated for dipolar bosons whose polarization direction is perpendicular to the plane of confinement.

Reviewer 2 Report

In the paper "Zero-temperature equation of state of a two-dimensional bosonic quantum fluid" the author uses a functional integral approach to derive the zero-temperature equation of state, extending previous results to the case of a finite range interaction.

This extensions is relevant to model more realistically a two-dimensional Bose gas, and will likely important in the correct description of future experiments.

The derivations are carried out in a very clear way, every step is carefully justified almost in a 'tutorial' style that will greatly help the readers.

The results presented are in an effective way, and the manuscript is carefully edited.

In summary, I think the paper should be accepted in its present form. I have just a couple of minor optional, stylistic suggestions for the author:

1) The Wick's rotation should be described as t  →  i τ, rather than t  =  i τ.

2) The title of the paper could be modified to emphasise the role of the finite-range part of the potential. 

Author Response

The author thanks the referee for accepting to referee the paper and for the kind appreciation on the style and the contents of the manuscript.

The referee writes: The Wick's rotation should be described as t  ?  i t, rather than t  =  i t.

Reply: The requested change has been made.

Referee writes: The title of the paper could be modified to emphasise the role of the finite-range part of the potential.

Reply: The author thanks the referee for the useful suggestion. The title has been changed to "Zero-temperature equation of state of a two-dimensional bosonic quantum fluid with finite-range interaction".